# The Association between Social Support Sources and Cognitive Function among Community-Dwelling Older Adults: A One-Year Prospective Study

**DOI:** 10.3390/ijerph16214228

**Published:** 2019-10-31

**Authors:** Taiji Noguchi, Ippei Nojima, Tomoe Inoue-Hirakawa, Hideshi Sugiura

**Affiliations:** 1Department of Social Science, Center for Gerontology and Social Science, National Center for Geriatrics and Gerontology, Aichi 474-8511, Japan; 2Department of Public Health, Nagoya City University Graduate School of Medical Sciences, Aichi 467-8601, Japan; 3Department of Health Sciences, Shinshu University Graduate School, Nagano 390-8621, Japan; 4Department of Physical Therapy, Graduate School of Medicine, Nagoya University, Aichi 461-8673, Japan

**Keywords:** cognitive function, community-dwelling older adults, prospective study, social support, Japanese

## Abstract

There is evidence that social relationships may modify cognitive decline in older people. We examined the prospective association between social support and cognitive function among community-dwelling older people. Japanese adults recruited at health checkups in suburban towns were surveyed at baseline and one-year follow-up. Cognitive function was assessed using the Montreal Cognitive Assessment, Japanese version (MoCA-J). Social support from coresiding family, non-coresiding family, and neighbors/friends was assessed using self-administered questionnaires. Multivariable linear regression analysis was conducted to examine the effects of social support on MoCA-J scores at follow-up. Data were analyzed from 121 older people (mean age (standard deviation): 73.86 (4.95) years). There was a positive association between social support exchanges with neighbors and friends and MoCA-J scores at follow-up after covariate adjustment (unstandardized β = 1.23, *p* = 0.006). Social support exchanges with coresiding family and non-coresiding family and relatives were not associated with MoCA-J scores at follow-up (coresiding family: Unstandardized β = 0.28, *p* = 0.813, non-coresiding family and relatives: Unstandardized β = 0.51, *p* = 0.238). The provision of emotional support to neighbors and friends had the largest effect on MoCA-J scores. Our findings suggest that social support exchanges with neighbors and friends are protective against cognitive decline.

## 1. Introduction

Dementia is a global health problem, as the number of people with dementia worldwide may exceed 82 million by 2030 [1]. Dementia and the decline of cognitive function greatly contribute to disability, health problems, and social care needs among older people [2]. Because there is no current cure for dementia, there is increasing interest in the identification of modifiable factors to delay or prevent dementia onset and the decline of cognitive function among older people [3].

According to the World Health Organization’s (WHO) policy framework on active aging, social relationships are a key factor in achieving active aging [4] and a potentially important modifiable factor that can decrease cognitive decline [3,5,6,7,8,9,10,11,12,13]. Several epidemiological studies have suggested that rich social relationships are protective against cognitive decline and the onset of dementia [7,8,9,10,11,12,14]. Rich social relationships may increase cognitive activity and prevent low mood, which is linked to delays in cognitive decline [6]. Studies of social relationships and health have used various measurement strategies to assess social relationships. One type of strategy uses a quantity-based approach to measure factors such as the number of social ties or the amount of participation in organizations. The other type of strategy uses a quality-based approach to measure characteristics such as the nature of social support.

Social support can be defined as an exchange between persons and is embedded in people’s social networks [15,16]. Social support is exchanged in the form of daily assistance, care, financial assistance, gift-giving, counseling, or emotional assurance [16]. Previous studies have suggested that social support has protective effects against cognitive decline and dementia onset [7,8,9,10,14,17]. The health and psychological impact of social support may vary according to the source of support. For instance, social support from friends may contribute to net subjective well-being, because friends are typically about the same age and often share similar cohort experiences, lifestyles, and residential proximity [18]. Support within the family, particularly from spouses, is often the main source of support in old age and has positive effects, but there may be gender differences in support [18]. Additionally, support from adult children is not necessarily positive, as relationships with children are often stressful and sometimes fraught with unwelcome reciprocal obligations [18]. Although it is important to understand differences in the effects of social support by differentiating between support sources inside (coresiding family) and outside (non-coresiding family and relatives, neighbors, and friends) the family, few studies have examined these social support sources separately.

A few previous studies have shown that different sources of social support have different effects. One cross-sectional study reported that family support was protectively associated with cognitive function among older people, but support from friends was not [12]. Another study found that support from friends had a positive effect on cognitive function [17]. In a longitudinal study, Murata et al. reported that social support from family, but not from relatives and friends, was protective against cognitive decline [10]. In contrast, Chen and Chang found that emotional support from friends or relatives had a protective effect [11]. There is little consensus in these findings owing to the small number of reports. Therefore, we aimed to examine the prospective effect of different sources of social support on cognitive function among community-dwelling older people through a one-year follow-up.

## 2. Materials and Methods

### 2.1. Study Population

In this prospective cohort study, baseline data were used from the Togo town study, which was carried out in cooperation with Nagoya University (Department of Health Sciences) and Togo town office. It was a community-based health-check survey conducted in July and August 2017. Individuals were selected for participation from people living in Togo town, a city in Aichi, Japan, and were recruited from a health-check project held in Togo town. A total of 284 adults were enrolled in the study. Baseline exclusion criteria were age < 65 years (*n* = 33), a history of diagnosis of dementia or mental illness (*n* = 5), non-independent walking ability (*n* = 4), and missing data for the cognitive function item (*n* = 4). The remaining 241 participants were followed for one year from baseline assessment and underwent cognitive testing again at the same center. Participants who did not complete a cognitive function survey at one year of follow-up (*n* = 120) were excluded. Therefore, 121 older adults met the criteria and were available for the final analysis.

Informed consent was obtained from all participants before their inclusion in the study, and the ethics committee of Nagoya University approved the study protocol (No. 18-502). The study was conducted in accordance with the guidelines of the Declaration of Helsinki.

### 2.2. Outcome measure: Cognitive Function

We assessed cognitive function at baseline and at follow-up using the Japanese version of the Montreal Cognitive Assessment (MoCA-J) [19,20]. The MoCA-J is a useful test that screens for mild cognitive impairment and is recommended for use in community-based geriatric health screenings [21]. The MoCA-J assesses six cognitive domains: Memory, visuospatial abilities, executive function, attention, concentration, and working memory, language, and time and place orientations. The total MoCA-J score ranges from 0 to 30. Higher scores indicate higher cognitive function. To correct for the effect of education, we added one point for participants with a total MoCA-J score < 30 points and educational attainment ≤ 12 years. To prevent measurement bias, the individuals who performed the assessment were trained to use the MoCA-J through lectures and role-playing prior to assessment.

### 2.3. Exposure Measure: Social Support

To assess social support, participants completed a self-administered questionnaire at baseline. This comprised the Two-Way Social Support Scale [22]. Four types of social support exchange were measured: Emotional support (providing and receiving) and instrumental support (providing and receiving). An example of emotional support is listening to someone’s concerns and complaints. An example of instrumental support is looking after someone for a few days when they are sick in bed. Participants were instructed to respond yes or no regarding support exchanges with the following: Spouse, children living together, children or relatives living separately, neighbors, and friends. We categorized three sources of social support: coresiding family (spouse and children living together), non-coresiding family and relatives, and neighbors and friends. For each of the three sources of social support, any support exchange was recorded as the presence of that type of support (the amount of support was not specified). These types of social support have frequently been assessed in previous studies [10,15,16].

### 2.4. Covariates

Sociodemographic characteristics, physical and psychological function, and healthy behaviors at baseline were assessed using a self-administered questionnaire and were included in the analyses as covariates: Age, sex, body mass index (BMI), living alone, equivalent income, medical history (stroke, hypertension, dyslipidemia, and diabetes, which show a relationship with cognitive decline) [3], depression, instrumental activities of daily living (IADL), walking speed, and walking time. BMI was calculated from height and weight measured using a multifrequency bioelectrical impedance analyzer (MC-780A, Tanita, Tokyo, Japan). Depression was assessed using the 15-item Japanese version of the Geriatric Depression Scale (GDS) [23]. Scores were used as a continuous variable (0 to 15 points, higher scores are associated with greater depression). IADL was measured using the five-item Tokyo Metropolitan Institute of Gerontology Index of Competence (TMIG-IC), which is based on the Lawton IADL scale [24]. Scores were used as a continuous variable (0 to 5, lower scores are associated with lower IADL ability). Walking speed was measured using a stopwatch. Participants were asked to walk on the flat and straight surface of a 10-m walk path at a comfortable walking speed (m/s). Walking time was measured to assess daily physical activity. Participants were asked how many days and for how long they walked for more than 10 min in a week, and we calculated daily walking time (hour/day).

### 2.5. Statistical Analysis

We first calculated descriptive statistics to compare dropout and non-dropout participants. Continuous variables (age, BMI, equivalent income, IADL score, GDS score, walking speed, walking time, and MoCA-J score) were analyzed using an unpaired t-test. Categorical variables (sex, living alone, medical history, and social support) were analyzed using the chi-squared test. Next, to examine the prospective association between cognitive function and the three types of social support (coresiding family, non-coresiding family and relatives, and neighbors and friends), we conducted multivariable linear regression analysis to obtain unstandardized regression coefficients (βs) and standard errors (SEs) for MoCA-J scores at follow-up. First, we conducted an analysis adjusted by age, sex, and MoCA-J score at baseline (model 1). Then we added BMI, living alone, equivalent income, medical history, GDS score, IADL score, walking speed, and walking time (model 2). In addition, we entered each social support item (receive/provide emotional support and receive/provide instrumental support) into the analysis model to examine the impact of each item on MoCA-J scores at follow-up.

To mitigate potential biases caused by missing information, we used the multiple imputation approach with the missing at random assumption (i.e., the missing data mechanism depends only on observed variables). We generated 20 imputed data sets using the multiple imputation by chained equations (MICE) procedure and pooled the results using the standard Rubin’s rule [25].

The significance level was set at *p* < 0.05. We used R version 3.4.3 for Windows R (Foundation for Statistical Computing, Vienna, Austria) for all statistical analyses. The multiple imputation approach involved the use of the MICE function (MICE package).

## 3. Results

Data from 121 participants were included in the analysis. The characteristics of both dropout and non-dropout participants are shown in Table 1. The mean age of the non-dropout participants was 73.86 years (standard deviation (SD) = 4.95 years), and the mean MoCA-J score was 25.98 (SD = 2.75). There was no significant difference in baseline data for the dropout and non-dropout participants, except for MoCA-J score. MoCA-J scores for the non-dropout participants were higher than those of the dropout participants (mean: 25.98, SD: 2.75 for non-dropout participants, 25.01, 3.42 for dropout participants, *p* = 0.015).

Table 2 shows the association between social support at baseline and MoCA-J score at follow-up. In model 1, the linear regression analysis adjusted by age, sex, and MoCA-J score at baseline revealed a positive association between exchange of social support with neighbors and friends and MoCA-J score at follow-up (unstandardized β = 1.10, SE = 0.41, *p* = 0.009). There was no significant association between social support exchanges with coresiding family and with non-coresiding family and relatives and MoCA-J score at follow-up (with coresiding family: Unstandardized β = −0.24, SE = 0.54, *p* = 0.665, non-coresiding family and relatives: Unstandardized β = 0.61, SE = 0.41, *p* = 0.138). In model 2, the association between exchange of social support with neighbors and friends and MoCA-J score at follow-up was also significant, adjusted for all covariates (unstandardized β = 1.23, SE = 0.44, *p* = 0.006). The association between social support from other sources and MoCA-J score at follow-up was not significant (with coresiding family: Unstandardized β = 0.28, SE = 1.19, *p* = 0.813, non-coresiding family and relatives: Unstandardized β = 0.51, SE = 0.43, *p* = 0.238).

To further examine the exchange of social support with neighbors and friends, we examined which subcomponents were strongly associated with MoCA-J scores at follow-up. Table 3 shows the results of a linear regression analysis using the subcomponents of social support exchanges with neighbors and friends as explanatory variables. The results showed that providing emotional support had a significant positive association with MoCA-J score at follow-up (unstandardized β = 1.25, SE = 0.56, *p* = 0.029). Other subcomponents of social support exchanges with neighbors and friends were not significantly related to MoCA-J scores (receiving emotional support: Unstandardized β = −0.42, SE = 0.57, *p* = 0.462, receiving instrumental support: Unstandardized β = 0.38, SE = 0.82, *p* = 0.642, providing instrumental support: Unstandardized β = 0.04, SE = 0.66, *p* = 0.948).

## 4. Discussion

This study investigated the prospective association between social support sources and cognitive function in community-dwelling older people through a one-year follow-up. We found that social support exchanges with neighbors and friends were positively associated with cognitive function at one-year follow-up. Regarding the type of social support from neighbors and friends, the provision of emotional support had a significant impact on the maintenance of cognitive function. Our findings suggest the importance of sources of social support in the community, such as neighbors and friends, for maintaining cognitive function among older people.

The findings showed that the exchange of social support with neighbors and friends was positively associated with cognitive function at follow-up. Some previous studies have reported that social support outside the family is positively associated with cognitive function [11,17]. In a study by Yeh, perceived positive support from friends (i.e., whether or not participants had a good friend to talk to) was associated with good cognitive function [17]. In Chen and Chang’s longitudinal study, social support outside the family was protective against the decline of cognitive function [11]. Individuals with social support outside the family, such as neighbors and friends, may experience rich social relationships and/or engage in social activities in the community, which may promote cognitive stimulation and delay cognitive decline [26]. In particular, the exchange of social support with friends may contribute to net psychological health because friends are typically about the same age and often share similar cohort experiences, lifestyle, and residential proximity [18]. In addition, social support from neighbors and friends has a positive impact on cognitive function, mediating avoidance of loneliness [14]. Social support outside the family might help to reduce loneliness. We believe that rich social exchanges with neighbors and friends may be important to maintain cognitive function.

Providing emotional support to neighbors and friends had a significantly positive effect on cognitive function after adjustment for all covariates. Some previous studies have also shown that emotional support is protective of cognitive function [7,11,14]. A prospective study showed that providing emotional support to friends is protective against dementia onset in men [10]. Our results are mostly consistent with this trend, although gender differences could not be considered because of the small sample size. Another cross-sectional study reported that the provision of emotional support to individuals outside the family was inversely associated with depression [27]. Prosocial attitudes and behaviors, informal helping, and altruistic attitudes may facilitate good mental health [28], which may have a positive effect on cognitive function. Our findings suggest the importance of opportunities to provide emotional support in the community, such as listening to others’ stories and problems.

Our results indicate that social support from both coresiding and non-coresiding family members was not significantly related to cognitive function. These findings are inconsistent with those of previous studies indicating that social support from family is protective against the onset of dementia or cognitive function decline [8,10,12]. As our study focused on individuals who were able to attend the test sitting at the local town hall, the health status of participants may have differed from the health status of participants in previous studies. Our participants may have been healthier or more socially active than those in previous studies. Social support within the family may not have a substantial effect on healthy older people. In other words, the lack of association between social support from non-coresiding family/relatives and cognitive function was consistent with previous research [10]. In recent years, household size has been shrinking in Japan owing to a rapid decline in traditional family systems. As in Western nations, adults are now more likely to live separately from their parents once they marry [29]. These changes in Japanese society may increase the importance of more diverse social networks (which include non-family members) among older adults. Therefore, as interaction with non-coresiding family, such as relatives and children living separately, is infrequent, the presence of support from non-coresiding family may indicate that an individual has poor health or a small decline in cognitive function and thus has potential care needs.

The present study has several strengths. First, the prospective study design allowed the identification of a possible causal effect of social support on cognitive function. Second, cognitive function was assessed using psychological tests for which validity has been confirmed. However, several study limitations should be considered. First, participants were recruited from individuals participating in a health checkup held in a suburban town hall. The participants were somewhat healthier and younger than typical community-dwelling older people living in the town. This may reduce the generalizability of our results. Second, there may have been selection bias. The follow-up rate was quite low (50.2%). Cognitive function at baseline for dropout participants was lower than that of non-dropout participants (MoCA-J score dropout participants: Mean = 25.01, SD = 3.42, non-dropout participants: Mean = 25.98, SD = 2.75, *p* = 0.015). However, there were no significant differences between non-dropout and dropout participants on the other variables, including social support. Therefore, we believe that the effects of dropout on the association between social support and cognitive function at follow-up may be relatively small. Third, the follow-up period was relatively short. The possibility of reverse causality in the association between social support and cognitive function owing to the effects of unidentified variables on the data cannot be ruled out. Further studies using long-term follow-up periods are needed. Fourth, the sample size was quite small. This did not permit the examination of gender differences using stratification analysis, although research suggests gender-related differences in the effects of social support [10]. We confirmed that there was no interactive effect of social support on cognitive function in this study group. Further studies are needed to examine gender differences using a large population. Fifth, social support was assessed using self-report questionnaires, which may have led to measurement error. However, the self-report questionnaire is the most frequently used tool to assess social support. Finally, we were unable to evaluate the impact of the amount of support on cognitive function because we only assessed the presence/absence of social support. Further studies that incorporate measures of the amount of support are needed.

Despite these limitations, the findings provide evidence of the prospective association between social support and cognitive function. As the global population ages, the number of older people living alone without family support is increasing [4]. As family resources become exhausted, the risks of adverse health outcomes increase [30,31]. According to the WHO report on active ageing, social support is a key social environmental factor that can enhance health, participation, and security as people age [4]. Our findings support the WHO report’s emphasis on social support. In Japan, a community-based integrated care system is being promoted to support older people to live independently in the community [32]. The importance of mutual support, such as exchanges of social support with neighbors and friends, is being highlighted in this system [32]. Policymakers and social service professionals can help community development that fosters social support exchanges for older people by supporting community groups, voluntarism, and neighborhood helping.

## 5. Conclusions

In conclusion, this prospective study showed that social support exchanges with neighbors and friends are positively associated with cognitive function among community-dwelling older people. Regarding supportive exchanges with neighbors and friends, providing emotional support had the largest effect on maintaining cognitive function. Our findings suggest the importance of social support exchanges in the community, including those with neighbors and friends, for maintaining cognitive function among older people.

## Figures and Tables

**Table 1 ijerph-16-04228-t001:** Participant characteristics.

Variable	Category	Dropout Participants (*n* = 120)	Non-Dropout Participants (*n* = 121)	*p*-Value *
Age (years)		73.78 (5.47)	73.86 (4.95)	0.900
Sex, *n* (%)	Men	65 (54.2)	64 (52.9)	0.897
	Women	55 (45.8)	57 (47.1)	
BMI (kg/cm^2^)		23.06 (3.10)	22.51 (3.03)	0.161
Living alone, *n* (%)	No	96 (80.0)	95 (78.5)	0.238
	Yes	11 (9.2)	18 (14.9)	
	Missing	13 (10.8)	8 (6.6)	
Equivalent income (10,000 JPY)		272.40 (163.09)	262.46 (123.73)	0.629
Medical history, *n* (%)				
Stroke	No	102 (85.0)	104 (86.0)	0.962
	Yes	11 (9.2)	11 (9.1)	
	Missing	7 (5.8)	6 (5.0)	
Hypertension	No	77 (64.2)	70 (57.9)	0.499
	Yes	36 (30.0)	45 (37.2)	
	Missing	7 (5.8)	6 (5.0)	
Dyslipidemia	No	85 (70.8)	82 (67.8)	0.807
	Yes	28 (23.3)	33 (27.3)	
	Missing	7 (5.8)	6 (5.0)	
Diabetes	No	96 (80.0)	104 (86.0)	0.444
	Yes	16 (13.3)	11 (9.1)	
	Missing	8 (6.7)	6 (5.0)	
IADL score		4.94 (0.27)	4.94 (0.27)	0.929
GDS score		2.50 (2.02)	2.22 (2.12)	0.358
Walking speed (m/s)		1.40 (0.20)	1.40 (0.23)	0.948
Walking time (hour/day)		0.88 (0.97)	0.86 (0.94)	0.864
Exchange of social support, *n* (%)				
Coresiding family	No	13 (10.8)	19 (15.7)	0.257
	Yes	100 (83.3)	99 (81.8)	
	Missing	7 (5.8)	3 (2.5)	
Non-coresiding family and relatives	No	40 (33.3)	42 (34.7)	0.475
	Yes	73 (60.8)	76 (62.8)	
	Missing	7 (5.8)	3 (2.5)	
Neighbors and friends	No	45 (37.5)	41 (33.9)	0.313
	Yes	68 (56.7)	77 (63.6)	
	Missing	7 (5.8)	3 (2.5)	
MoCA-J score at baseline		25.01 (3.42)	25.98 (2.75)	0.015
MoCA-J score at follow-up		-	26.11 (3.00)	-

Age, BMI, equivalent income, IADL score, GDS score, walking speed, walking time, and MoCA-J scores are expressed as the mean (standard deviation). Other variables are expressed as n (%). * Unpaired t-test for continuous variables and chi-squared test for categorical variables. For dropout participants, the total number of data points for equivalent income was 95, the IADL score was 106, and the GDS score was 90, owing to missing data. For non-dropout participants, the total number of data points for equivalent income was 102, the IADL score was 117, and the GDS score was 99 owing to missing data. BMI: Body mass index, GDS: Geriatric Depression Scale, IADL: Instrumental activities of daily living, JPY: Japanese yen, MoCA-J: Montreal Cognitive Assessment, Japanese version.

**Table 2 ijerph-16-04228-t002:** The effects of social support on cognitive function at the one-year follow-up.

	Model 1		Model 2	
Variable	Unstandardized β (SE)	*p*-Value	Unstandardized β (SE)	*p*-Value
Exchange of social support				
Coresiding family	−0.24 (0.54)	0.665	0.28 (1.19)	0.813
Non-coresiding family and relatives	0.61 (0.41)	0.138	0.51 (0.43)	0.238
Neighbors and friends	1.10 (0.41)	0.009	1.23 (0.44)	0.006
Age (years)	−0.14 (0.04)	0.003	−0.13 (0.05)	0.008
Women	−0.77 (0.42)	0.068	−0.75 (0.45)	0.093
MoCA-J score at baseline	0.62 (0.08)	<0.001	0.59 (0.08)	< 0.001
BMI (kg/cm^2^)			0.06 (0.07)	0.384
Living alone			0.48 (1.22)	0.693
Equivalent income (per 1,000,000 JPY)			−0.13 (0.19)	0.485
Stroke			−0.42 (0.70)	0.548
Hypertension			−0.07 (0.44)	0.871
Dyslipidemia			0.29 (0.48)	0.544
Diabetes			−1.80 (0.72)	0.014
IADL score			−0.78 (0.70)	0.268
GDS score			−0.06 (0.10)	0.553
Walking speed (m/s)			0.73 (0.88)	0.405
Walking time (hour/day)			−0.05 (0.21)	0.818

BMI: Body mass index, GDS: Geriatric Depression Scale, IADL: Instrumental activities of daily living, JPY: Japanese yen, MoCA-J: Montreal Cognitive Assessment, Japanese version, SE: Standard error.

**Table 3 ijerph-16-04228-t003:** The effects of subcomponents of social support exchanges with neighbors and friends on cognitive function at one-year follow-up.

Type of Social Support	Unstandardized β (SE)	*p*-value
Social support exchanges with neighbors and friends		
Receiving emotional support	−0.42 (0.57)	0.462
Providing emotional support	1.25 (0.56)	0.029
Receiving instrumental support	0.38 (0.82)	0.642
Providing instrumental support	0.04 (0.66)	0.948

Adjusted for age, sex, BMI, living alone, equivalent income, medical history, IADL, depression, walking speed, walking time, social support exchanges with coresiding family, social support exchanges with non-coresiding family and relatives, and MoCA-J score at baseline. SE: Standard error, BMI: Body mass index, IADL: Instrumental activities of daily living, MoCA-J: Montreal Cognitive Assessment, Japanese version.

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
