# Peer review of "The Association between Social Support Sources and Cognitive Function among Community-Dwelling Older Adults: A One-Year Prospective Study"

_ijerph, 2019, doi:10.3390/ijerph16214228_

Round 1

Reviewer 1 Report

A good study exploring the link between social support and ageing, adding value to the literature regarding how social connection in later life is supportive. 

The study limitations have been adequately explored and explained, including potential for reverse causality. 

Author Response

Reviewer 1

Comment #1

A good study exploring the link between social support and ageing, adding value to the literature regarding how social connection in later life is supportive.

The study limitations have been adequately explored and explained, including potential for reverse causality.

Response #1

Thank you for your comments. We believe that it is necessary for Japan, which has the highest ageing rate in the world, to send out messages that can contribute to healthy ageing. There were some study limitations, but we hope that our findings will be useful for future research on healthy ageing.

Reviewer 2 Report

Overall, the paper reads well and makes significant contribution to the ageing literature.

May main issue is that the authors did not situate the paper in the right theoretical and policy contexts.

My suggestion would be for the authors to place the paper within the context of of the 2002 World Health Organisation (WHO) proposed active ageing policy framework which guides countries to develop policies that promote the quality of life for their older people.

In my opinion, providing this context would the paper an international appeal and significance.

The authors may also wish to look at the findings of Dr Daniel Doh's thesis "Towards active ageing: A comparative study of experiences of older Ghanaians in Australia and Ghana"

Author Response

Reviewer 2

Comment #1

Overall, the paper reads well and makes significant contribution to the ageing literature. May main issue is that the authors did not situate the paper in the right theoretical and policy contexts. My suggestion would be for the authors to place the paper within the context of of the 2002 World Health Organisation (WHO) proposed active ageing policy framework which guides countries to develop policies that promote the quality of life for their older people. In my opinion, providing this context would the paper an international appeal and significance. The authors may also wish to look at the findings of Dr Daniel Doh's thesis "Towards active ageing: A comparative study of experiences of older Ghanaians in Australia and Ghana"

Response #2

Thank you for these important insights. We agree with your suggestions. The WHO’s active ageing policy framework emphasizes the social environment, including fostering social support, as a key factor in promoting the health and well-being of older people. Our findings support the recommendation of this report. We have revised our discussion of policymaking in accordance with your suggestions (pp. 1, lines 42–43): “According to the World Health Organization’s (WHO) policy framework on active ageing, social relationships are a key factor in achieving active ageing [4]”; (pp. 8, lines 276–277): “As the global population ages, the number of older people living alone without family support is increasing [4]”; (pp. 8, lines 278–280): “According to the WHO report on active ageing, social support is a key social environmental factor that can enhance health, participation, and security as people age [4].Our findings support the WHO report’s emphasis on social support.”; (pp. 8, lines 283–285): “Policymakers and social service professionals can help community development that fosters social support exchanges for older people by supporting community groups, voluntarism, and neighborhood helping.”

Reviewer 3 Report

I read the manuscript with great interest. The manuscript is generally well written. However, I have some methodological remarks:

The authors assessed the baseline cognitive status and at 1-year follow-up. However, the statistical analysis regrading the association of different variables including social support was performed on the MOCA score at follow-up. There is no information regarding the change in the MOCA score with time and no analysis was performed using the association of the change in MOCA score and social support. Social support was assessed using a questionnaire at baseline that measures three sources of social support. It is not clear how the amount of social support was measured did they use any validated scale (there are a few validated scales measuring social support) or did they only ask a yes/no question about the presence of each type of social support? According to the methods section they decided to define any support exchange as the presence of that type of support, so the amount of each type of support was probably not recorded. The authors should explain this methodological decision.

Other comments

The finding that providing emotional support to neighbors and friend was the component of social support that was associated with cognitive function is interesting. The authors discussed the influence of the Japanese culture on the interaction with family members (lines 240-243). Since receiving and providing social support is culturally dependent it would be interesting if the authors elaborate more on the possible influence of culture on the relationships with family members and with neighbors and friends and compare to findings in other cultures.

Author Response

Reviewer 3

Comment #1

I read the manuscript with great interest. The manuscript is generally well written. However, I have some methodological remarks: The authors assessed the baseline cognitive status and at 1-year follow-up. However, the statistical analysis regrading the association of different variables including social support was performed on the MOCA score at follow-up. There is no information regarding the change in the MOCA score with time and no analysis was performed using the association of the change in MOCA score and social support.

Response #1

Thank you for your comments. To examine the longitudinal impact of social support on cognitive function (MoCA-J score), we performed regression analysis with MoCA-J score at follow-up as the dependent variable, social support at baseline as the explanatory variable, and MoCA-J score at baseline as the covariate. As you point out, we could have analyzed the relationship between changes in MoCA-J score and social support at baseline. However, the analysis model with changes in the MoCA-J score as the dependent variable assumes a regression coefficient (β) of β = 1 for the MoCA-J score at baseline. As we felt that this assumption was relatively strong, we did not perform this analysis. Our analysis showed that the β of the MoCA-J score at baseline was 0.59, which is less than β = 1. We also chose this approach because we wanted to examine the longitudinal association with social support rather than changes in the MoCA-J score.

              However, we did conduct an analysis (data not shown in the manuscript) to confirm the changes in the MoCA-J score. This showed that the exchange of social support with neighbors and friend was positively associated with the change in MoCA-J score, which showed almost the same pattern as the reported results (social support with neighbors and friend: β = 1.13, SE = 0.49, p = 0.021; coresiding family: β = 0.14, SE = 1.31, p = 0.910; non-coresiding family and relatives: β = 0.20, SE = 0.48, p = 0.671).

Comment #2

Social support was assessed using a questionnaire at baseline that measures three sources of social support. It is not clear how the amount of social support was measured did they use any validated scale (there are a few validated scales measuring social support) or did they only ask a yes/no question about the presence of each type of social support? According to the methods section they decided to define any support exchange as the presence of that type of support, so the amount of each type of support was probably not recorded. The authors should explain this methodological decision.

Response #2

Thank you for your important questions. We assessed social support using the 2-Way Social Support Scale, which has demonstrated factor, convergent, and predictive validity (Shakespeare-Finch J and Obst PL, J Pers Assess. 2011). We used this scale because it has been used in a nationwide survey of older people conducted by the Japanese Ministry of Health, Labour and Welfare to formulate a public long-term care insurance plan (https://www.mhlw.go.jp/stf/shingi2/0000138653.html, accessed 19 October 2019). Based on previous study findings, four types of social support were assessed: emotional support (providing and receiving) and instrumental support (providing and receiving). We assessed five support sources for each of these social support types: spouse, children living together, children or relatives living separately, neighbors, and friends. We categorized three sources of social support: coresiding family (spouse and children living together), non-coresiding family and relatives, and neighbors and friends, on the basis of a previous study (Murata C. et al., Int J Environ Res Public Health. 2017; Murata C. et al., Int J Environ Res Public Health. 2019). However, because any support exchange was recorded as the presence of that type of support for the three sources, we were unable to assess the effect of the amount of social support. We have revised the description of how social support was assessed in the Materials and Methods (pp. 3, lines 104–114): “This comprised the 2-Way Social Support Scale. [22] Four types of social support exchange were measured: emotional support (providing and receiving) and instrumental support (providing and receiving). An example of emotional support is listening to someone’s concerns and complaints; an example of instrumental support is looking after someone for a few days when they are sick in bed. Participants were instructed to respond yes or no regarding support exchanges with the following: spouse, children living together, children or relatives living separately, neighbors, and friends. We categorized three sources of social support: coresiding family (spouse and children living together), non-coresiding family and relatives, and neighbors and friends. For each of the three sources of social support, any support exchange was recorded as the presence of that type of support (the amount of support was not specified). These types of social support have frequently been assessed in previous studies [10,15,16].”

We have also mentioned in the discussion of the study limitations that we were unable to examine the impact of the amount of social support (Discussion, pp. 8, lines 272–274): “Finally, we were unable to evaluate the impact of the amount of support on cognitive function because we only assessed the presence/absence of social support. Further studies that incorporate measures of the amount of support are needed.”

Comment #3

Other comments

The finding that providing emotional support to neighbors and friend was the component of social support that was associated with cognitive function is interesting. The authors discussed the influence of the Japanese culture on the interaction with family members (lines 240-243). Since receiving and providing social support is culturally dependent it would be interesting if the authors elaborate more on the possible influence of culture on the relationships with family members and with neighbors and friends and compare to findings in other cultures.

Response #3

Thank you for your helpful advice. As you pointed out, social support exchange is closely related to culture, and may reflect differences between individualist Western countries and collectivist Asian countries such as Japan. Because the cost of seeking help from others, especially from non-family members (e.g., neighbors or friends) might be higher in Asian countries than in Western countries, social support from family members such as spouses and children may be more important to health among Japanese older adults. However, owing to a rapid decline in traditional family systems in recent years, these cultural differences are perhaps becoming less relevant. Household size in Japan is shrinking, and more adults now live separately from their parents once they get married (Brown JW. et al., J Gerontol B Psychol Sci Soc Sci. 2002). We believe that these changes in Japanese society may increase the importance to older adults of more diverse social networks that include individuals outside the family, such as neighbors and friends. We have added a discussion of this issue to the revised manuscript (pp. 7, lines 243–249): “In recent years, household size has been shrinking in Japan owing to a rapid decline in traditional family systems; as in Western nations, adults are now more likely to live separately from their parents once they marry [29]. These changes in Japanese society may increase the importance of more diverse social networks (which include non-family members) among older adults. Therefore, as interaction with non-coresiding family, such as relatives and children living separately, is infrequent, the presence of support from non-coresiding family may indicate that an individual has poor health or a small decline in cognitive function and thus has potential care needs.”